# The Uterus as an Influencing Factor for Late Embryo/Early Fetal Loss—A Clinical Update

**DOI:** 10.3390/ani12151873

**Published:** 2022-07-22

**Authors:** Zoltán Szelényi, Levente Kovács, Ottó Szenci, Fernando Lopez-Gatius

**Affiliations:** 1Department of Obstetrics and Farm Animal Clinic, University of Veterinary Medicine, 1078 Budapest, Hungary; szenci.otto@univet.hu; 2Department of Animal Husbandry and Welfare, Hungarian University of Agricultural and Life Sciences, 2100 Gödöllő, Hungary; kovacs.levente@uni-mate.hu; 3Transfer in Bovine Reproduction SLu, 22300 Barbastro, Spain; lopezgatiusf@gmail.com

**Keywords:** age, cattle, nulliparous, pregnancy failure, reproductive management, retained fetal membranes, uterine involution

## Abstract

**Simple Summary:**

Pregnancy loss following a positive pregnancy diagnosis in the absence of infectious disease of the reproductive system is a main factor limiting reproductive efficiency in high producing dairy cows. We describe here some circumstances such as the age of the dam, retained placenta and uterine size in which the uterus may be associated with pregnancy loss.

**Abstract:**

Here we revise circumstances of non-infectious causes in which the uterus may be associated with pregnancy loss during the late embryo/early fetal period (following a positive pregnancy diagnosis in lactating dairy cows). As the uterine size increases with parity and pregnant heifers with no detrimental effects of a previous parturition, a primigravid uterus is proposed as a reference for identifying risk factors that negatively influence pregnancy in lactating cows. Cows suffering placenta retention or with a large uterus at insemination were selected as topics for this revision. Retained placenta, that occurs around parturition, has a long-lasting influence on subsequent pregnancy loss. Although retained placenta is a particularly predisposing factor for uterine infection, farm conditions along with cow factors of non-infectious cause and their interactions have been identified as main factors favoring this disorder. A large uterus (cervix and uterine horns lying outside the pelvic cavity) with no detectable abnormalities has been associated with low fertility and with a greater incidence of pregnancy loss. A large reproductive tract may well derive from an inadequate uterine involution. Therefore, peripartum management and strategies to reduce the incidence of uterine disorders should reduce their associated financial losses in the herds.

## 1. Introduction

Pregnancy loss in the absence of infectious disease of the reproductive system is a main factor limiting reproductive efficiency in high producing dairy cows. Following a positive pregnancy diagnosis 28–34 days post-artificial insemination (AI), pregnancy loss mainly occurs before day 60 of gestation [1,2,3,4]. Implantation is firmly established following this time interval [5,6] and the chances of losses are much lower [1,3,4]. The embryonic period of pregnancy spans from conception until the end of the differentiation stage (about 45 days), and the fetal period runs from day 45 of pregnancy to parturition [7]. Most losses occur in the early embryonic period, with the greatest increment of loss between days 15 and 18 [1]. However, losses during the late embryonic period can substantially exceed 20% [2,3,4], or even 40% [8,9] and cause a major economic impact in dairy herds [10,11]. The cost resulting from each cow suffering pregnancy failure can easily exceed $2000 due to an extended non-pregnant period and increased risk of culling [10]. Although an accurate pregnancy diagnosis can be performed on day 20 of gestation, using ultrasonography or conceptus protein determinations [4,12], pregnancy is routinely diagnosed on days 28–34 post-AI [4,13]. Numerous factors of non-infectious nature [1,2,3,4] and postpartum diseases [14,15,16,17] have been associated with pregnancy loss. Here we highlight circumstances of non-infectious cause in which the uterus may be associated with pregnancy loss during the late embryo/early fetal period in lactating dairy cows.

## 2. The Primigravid Uterus

The uterine size increases with parity [18,19], with dam age being a main factor associated with pregnancy loss [3,20,21]. Therefore, it should be expected that the primigravid uterus, with no detrimental effects of parturition and not exposed to the various metabolic and environmental stresses associated with the lactation period, may favor pregnancy maintenance in heifers when compared to parous cows. In effect, pregnant heifers show the lowest risk of pregnancy loss compared to lactating cows [22,23,24]. With registered losses of 9.6% (303/3162) for parous cows and 2.8% (29/1050) for heifers and using parous cows as reference, the odds ratio for the pregnancy loss in heifers was 0.28 (*p* = 0.0001) [22]. In a compilation of studies, 10.7% (663/6195) of losses were reported for lactating cows and 2.52% (84/3333) for primigravid heifers [24]. However, this difference may currently be masked with the use of sexed semen. The use of sex-sorted sperm, particularly for heifers and primiparous cows [25,26], has increased during the last two decades [27,28,29]. In this context, the use of sexed semen in heifers may increase the incidence of pregnancy loss [30]. This disadvantage to maintain a gestation derived from sexed semen should be assessed in extensive studies. While the refinement of the sex-sorting technology after 2015 has resulted in a better reproductive performance, the use of sexed semen leaves much scope for improvement [29]. Therefore, pregnant heifers following AI with conventional semen remain a good reference in studying factors influencing pregnancy loss in lactating cows.

## 3. Retained Placenta

The fetal membranes are physiologically expelled within two to eight hours after the delivery of the fetus [31], whereas retention of the entire or of parts of the membranes beyond 12 h is considered a pathological condition [31,32,33]. With a percentage of pregnancy loss of 9.6% (291/3022) in a study population with 6.3% (192/3022) cows suffering placenta retention and using cows with no retained placenta as reference, the odds ratio for the pregnancy loss in retained placenta cows was 1.8 (*p* = 0.005) [34]. The interval from previous parturition to pregnancy ranged from 30 to 496 days and could not be associated with the risk of pregnancy loss [29]. Although only clinically normal cows were inseminated, previous retained placenta could cause non-detected uterine lesions favoring subsequent pregnancy loss [29]. This suggests a long-lasting influence of previous retained placenta on pregnancy loss. In a more recent study, a risk factor for 126 herds having a high prevalence of pregnancy loss was ≥4.9% retained placenta [15].

The precise mechanisms that lead to placental release are not well understood. Expulsion of fetal membranes involves multiple hormonal, physical and immunological changes [35,36,37]. Although retained placenta is a particularly predisposing factor for uterine infection [38,39,40], farm conditions along with cow factors of non-infectious cause and their interactions have been identified as main factors favoring this disorder [41,42,43]. Therefore, strategies to reduce the incidence of retained placenta in the herds cannot be overemphasized. It is so critical that it is likely to determine how poorly the cow will perform during the lactation. The cost due to retained placenta has been estimated up to $481 and has been mainly associated with reduction in milk production [44]. In other words, financial consequences associated with placenta retention should be added to those pregnancy loss noted above.

## 4. Uterine Size

A negative association between uterine size and fertility has been described in lactating dairy cows [45,46,47]. A reproductive tract score system was developed to identify decreased fertility in lactating dairy cows with non-detectable reproductive disorders [46]. This system considers the size and position of the reproductive tract relative to the pelvis [46]. Briefly, cows undergoing pre-breeding exams using transrectal palpation were registered with a reproductive tract small (small uterine horns that rested within the pelvic cavity), medium (longer uterine horns resting partially outside the pelvic cavity), or large (larger cervix and uterine horns lying outside the pelvic cavity) [46]. Cows with large reproductive tracts had reduced fertility and a greater incidence of pregnancy loss between 31- and 60-days post-AI [47]. Pregnancy loss was increased in cows with a large reproductive tract compared to medium and small size (24.3% vs. 11.6 ± 0.02 and 9.4 ± 0.02%, respectively; *p* = 0.04) [47]. Primiparous cows had a higher frequency (*p* < 0.01) of small reproductive tract and a lower frequency of large reproductive tract when compared with multiparous cows (small: 42.6 vs. 15.0%; large: 7.0 vs. 22.0%, respectively) [47]. Interestingly, there was no interaction between parity and uterine size on pregnancy at AI [46,47]. This suggests that the reproductive tract score system may be a better indicator of low fertility than lactation number. Culling was associated among other factors with the uterine size [47]. Unfortunately, previous uterine disorders such as placenta retention or endometritis were not included in these studies. Further studies are necessary to determine factors influencing the development of a large uterus. A large reproductive tract may well derive from an inadequate uterine involution. Alterations in the uterine environment might undermine the delicate process of placentation predisposing to pregnancy loss. Moreover, the process of conceptus/maternal attachment between 36 and 47 days of gestation shows a high degree of individual variation [6]. This means that a large uterus can take much longer to complete placentation, a vulnerable process which would be exposed for a longer time to different types of stress.

## 5. Conclusions

A primigravid uterus may be a good reference in studying risk factors that negatively influence pregnancy in lactating cows.

Retained placenta has a long-lasting influence on subsequent pregnancy loss. Therefore, calving management and strategies to reduce the incidence of this uterine disorder will reduce its associated financial losses in the herds. 

Size and position of the uterus may provide a useful source of information to identify strategies to reduce pregnancy losses. 

## Data Availability

Not applicable.

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
