# Peer review of "The Uterus as an Influencing Factor for Late Embryo/Early Fetal Loss—A Clinical Update"

_animals, 2022, doi:10.3390/ani12151873_

Round 1
Reviewer 1 Report
The paper entitled “The uterus as an influencing factor for late embryo/early fetal loss. A clinical update” has an appealing title because this topic has great relevance. The paper discussed the relationship between age of the dam, incidence of retained placenta, and size of uterine horns with the probability of pregnancy losses. Although an update regarding these relationships is important, the authors could improve the quality of the paper by adding information regarding the physiological consequences that lead to the losses in the established pregnancy. The title calls for the importance of the uterus in the pregnancy loss, however, is missing in the text how the uterine environment/uterine physiology changes as a consequence of the age of the dam, retained placenta and size of the uterus, and how these changes contribute to a loss in pregnancy. Adding these arguments will improve the overall quality of the paper, and it will become more interesting for the readers. See some comments below:
Abstract
The abstract could be improved. In your title, it is stated the importance of the uterus for pregnancy loss. However, the abstract has only described the topics you choose are related to pregnancy loss, without explaining the reasons.
Line 15-16: Add the “)” at the end of the sentence.
Introduction
Line 29: The authors need to be careful about the sentences regarding the absence of infectious disease as a cause of pregnancy loss and including retention of placenta in this. Authors need to be clearer that retention of placenta occurs around calving, but at the breeding time, the probability of cows still having uterine infections is low. Please check https://doi.org/10.1016/j.tvjl.2007.12.031.
Line 31: Please be careful in affirming that the main period for pregnancy loss is between the pregnancy diagnosis (28-34 d after AI) to 60 d. Please check the following references: http://dx.doi.org/10.21451/1984-3143-AR2018-0028
https://doi.org/10.3168/jds.2017-13469
https://doi.org/10.1016/j.theriogenology.2016.04.037
The greater proportion of pregnancy losses occurs before the pregnancy diagnosis, mainly for failures in embryo elongation.
Line 39: Increased risk of culling?
Line 40: Please add the methodology for pregnancy diagnosis before Day 20 of gestation.
Line 41-24: I didn’t understand the sentence about twin pregnancy, and genomic prediction… This sentence does not add anything to the understanding of the introduction. Authors should consider removing it.
Introduction can be improved. Introduction should introduce your topics, the reasons why you believe the age of the dam, incidence of retained placenta, and size of uterine horns are important for pregnancy loss. Why should we care? These are the only reasons? Why only non-infectious causes?
The Primigravid Uterus
What are the reasons for heifers to have a lower incidence of pregnancy losses than multiparous cows? If the only reason is the size of the reproductive tract, it does not have a point to have this separate chapter, if you already had one for uterine size.
Line 54-56: Please rewrite the sentence, it is difficult to understand.
Line 59: In what stage of the pregnancy authors is referring? Explain the reasons for pregnancy loss using sexed semen? Sexed semen reduces fertilization rate, but regarding pregnancy loss, I believe there are controversial data. Please check https://doi.org/10.3390/ani10060925
Line 60: Carrying a pregnancy from sexed-semen is not a disadvantage. Although I understand that authors believed that using sexed-semen could increase the risk for pregnancy loss, the reason for the use of sexed-semen was revolutionary in the dairy industry. Please be careful with your statements.
Retained Placenta
Only the incidence of retained placenta is related to an increase in pregnancy losses? If retained placenta increases the likelihood of the occurrence of metritis, is metritis also related to pregnancy losses? Should authors consider talking about uterine diseases instead?
Line 69 – 71: Please rewrite this sentence, it is not clear.
Line 72-74: Is the interval from parturition to pregnancy not associated with pregnancy loss? Not clear again.
Line 74: Why do the authors believe retained placenta has a long-lasting effect on pregnancy loss? How could authors explain an increase in pregnancy loss as a consequence of retained placenta even after the cure of the clinical case? http://dx.doi.org/10.21451/1984-3143-AR1002
Line 77-86: Although it is interesting the mechanisms that lead to placental retention, the focus of your review is related to retained placenta with pregnancy loss. I suggest excluding this paragraph and focusing in answer the information you introduce in your title.
Uterine Size
Please include the hypothesis of how a larger uterine size negatively affects the maintenance of pregnancy.
Conclusion
After changing the text, the conclusion should be rewritten.
Author Response
Reviewer 1
The paper entitled “The uterus as an influencing factor for late embryo/early fetal loss. A clinical update” has an appealing title because this topic has great relevance. The paper discussed the relationship between age of the dam, incidence of retained placenta, and size of uterine horns with the probability of pregnancy losses. Although an update regarding these relationships is important, the authors could improve the quality of the paper by adding information regarding the physiological consequences that lead to the losses in the established pregnancy. The title calls for the importance of the uterus in the pregnancy loss, however, is missing in the text how the uterine environment/uterine physiology changes as a consequence of the age of the dam, retained placenta and size of the uterus, and how these changes contribute to a loss in pregnancy. Adding these arguments will improve the overall quality of the paper, and it will become more interesting for the readers.
(Au): Thank you very much for your consideration. We tried to add these arguments throughout the revised version.
Abstract
The abstract could be improved. In your title, it is stated the importance of the uterus for pregnancy loss. However, the abstract has only described the topics you choose are related to pregnancy loss, without explaining the reasons.
(Au) The reasons for these topics have been clarified. Please note that the abstract consists of no more than 200 words. With the revision we have now 197 words.
Line 15-16: Add the “)” at the end of the sentence.
(Au) Done.
Introduction
Line 29: The authors need to be careful about the sentences regarding the absence of infectious disease as a cause of pregnancy loss and including retention of placenta in this. Authors need to be clearer that retention of placenta occurs around calving, but at the breeding time, the probability of cows still having uterine infections is low. Please check https://doi.org/10.1016/j.tvjl.2007.12.031.
(Au) Thank you very much for your comment and reference. We agree. This point was emphasized in the submitted version with the proposed (40) and two further references in the subsection on retained placenta. This has now been clarified in the abstract.
Line 31: Please be careful in affirming that the main period for pregnancy loss is between the pregnancy diagnosis (28-34 d after AI) to 60 d. Please check the following references:
http://dx.doi.org/10.21451/1984-3143-AR2018-0028
https://doi.org/10.3168/jds.2017-13469
https://doi.org/10.1016/j.theriogenology.2016.04.037
The greater proportion of pregnancy losses occurs before the pregnancy diagnosis, mainly for failures in embryo elongation.
(Au) Thank you again for the references and comments. We were talking about losses following a positive pregnancy diagnosis. Anyway, a sentence on early embryo losses has been added.
Line 39: Increased risk of culling?
(Au) Corrected.
Line 40: Please add the methodology for pregnancy diagnosis before Day 20 of gestation.
(Au) Done.
Line 41-42: I didn’t understand the sentence about twin pregnancy, and genomic prediction… This sentence does not add anything to the understanding of the introduction. Authors should consider removing it.
(Au) The sentence and their corresponding references have been deleted.
Introduction can be improved. Introduction should introduce your topics, the reasons why you believe the age of the dam, incidence of retained placenta, and size of uterine horns are important for pregnancy loss. Why should we care? These are the only reasons? Why only non-infectious causes?
(Au) This is not an extensive review, just a commentary. We consider that the reasons to include the selected topics are better explained in their corresponding sub-sections. The focus has been clarified in the introduction and the importance of postpartum diseases has also been included.
The Primigravid Uterus
What are the reasons for heifers to have a lower incidence of pregnancy losses than multiparous cows? If the only reason is the size of the reproductive tract, it does not have a point to have this separate chapter, if you already had one for uterine size.
(Au) The sentence has been expanded: “… the primigravid uterus, with no detrimental effects of parturition and not exposed to the various metabolic and environmental stresses associated with the lactation period, may favor pregnancy maintenance…”
Line 54-56: Please rewrite the sentence, it is difficult to understand.
(Au) The sentence has been rewritten.
Line 59: In what stage of the pregnancy authors is referring? Explain the reasons for pregnancy loss using sexed semen? Sexed semen reduces fertilization rate, but regarding pregnancy loss, I believe there are controversial data. Please check https://doi.org/10.3390/ani10060925
Line 60: Carrying a pregnancy from sexed-semen is not a disadvantage. Although I understand that authors believed that using sexed-semen could increase the risk for pregnancy loss, the reason for the use of sexed-semen was revolutionary in the dairy industry. Please be careful with your statements.
(Au) The end of the sentence in Line 60 (“to similar numbers to those lactating cows”: current Line 74) has been deleted. We agree, the use of sexed semen was revolutionary, but as indicated, “leaves yet much scope for improvement [29]”. We just propose pregnant heifers with conventional semen as a reference in studying predisposing factors to pregnancy loss.
Retained Placenta
Only the incidence of retained placenta is related to an increase in pregnancy losses? If retained placenta increases the likelihood of the occurrence of metritis, is metritis also related to pregnancy losses? Should authors consider talking about uterine diseases instead?
(Au) Some references on relationships between postpartum diseases and pregnancy loss have been added in the introduction. We emphasized the fact that “farm conditions along with cow factors of non-infectious cause and their interactions have been identified as main factors favoring this disorder [42–44]”. Therefore, from a clinical point of view, reduction of the incidence of placenta retention is proposed as a clinical improvement in the herds.
Line 69 – 71: Please rewrite this sentence, it is not clear.
Line 72-74: Is the interval from parturition to pregnancy not associated with pregnancy loss? Not clear again.
(Au) Both sentences have been rewritten.
Line 74: Why do the authors believe retained placenta has a long-lasting effect on pregnancy loss? How could authors explain an increase in pregnancy loss as a consequence of retained placenta even after the cure of the clinical case? http://dx.doi.org/10.21451/1984-3143-AR1002
(Au) It has been clarified.
Line 77-86: Although it is interesting the mechanisms that lead to placental retention, the focus of your review is related to retained placenta with pregnancy loss. I suggest excluding this paragraph and focusing in answer the information you introduce in your title.
(Au) As noted above in our first response to “retained placenta”, we think that this is an explanatory key for this sub-section: “Although retained placenta is a particularly predisposing factor for uterine infection [39–41], farm conditions along with cow factors of non-infectious cause and their interactions have been identified as main factors favoring this disorder [42–44]”
Uterine Size
Please include the hypothesis of how a larger uterine size negatively affects the maintenance of pregnancy.
(Au) Two hypotheses have been added.
Conclusion
After changing the text, the conclusion should be rewritten
(Au) The part on uterine size has been rewritten.
Thank you very much for your perceptive and constructive criticism.
Reviewer 2 Report
Loss of pregnancy is a major factor limiting reproductive performance in high-yielding dairy cows. In recent years, both early embryonic death, as well as infectious or non-infectious abortions in cattle have been the subject of extensive research.
In this meaning, the presented for evaluation manuscrypt describing the most important non-infectious causes of pregnancy loss is very timely and needed.The Authors describe here some circumstances such as the age of the dam, retained placenta and uterine size in which the uterus may be associated with pregnancy loss.
The manuscript is well written and adequate references are cited. I have no critical comments.
Author Response
Authors: Thank you very much for your consideration.
Reviewer 3 Report
The authors commented relationship between pregnancy loss and primigravid gestation, retained placenta, and uterine size. This is an interesting topic, but the authors do not have a previous study on the subject and the article does not contain up-to-date and innovative information. Therefore, it is not appropriate to publish the manuscript.
Author Response
Authors: Thank you very much for your revision and comments. It is true, although we are using this score in the commercial dairy herds under our surveillance, we have not a previous study on uterine score and pregnancy loss. However, we have numerous studies on pregnancy loss during the late embryonic/early fetal period, some of them including pregnant heifers and retained placenta as factors. For example, the cited reviews [3] and [12].
Round 2
Reviewer 1 Report
The authors corrected the paper as suggested, and it improved the quality of the manuscript. I am still a little concerned about the retained placenta listed as a cause of non-infectious disease. Although I understand that the authors are affirming that the CAUSES of retantion of the placenta are related to farm conditions and cow factors, however, the CONSEQUENCE of retained placenta is an increased concentration of microorganisms in the uterus (see reference https://doi.org/10.3168/jds.2016-11623), which consequently increase the likelihood of cows to develop metritis, and could help explain why those cows have worst reproductive performance than cows that did not develop uterine disease after calving.
The conclusion is too long and could be improved to only conclude the ideas the authors introduce throughout the text.
Author Response
Please find authors reply in the attachment.

Reviewer 3 Report
It is not appropriate to publish the manuscript.
Author Response

(The authors gave the same response as above.)
